# Synthesis of Nano-Structured Ge as Transmissive or Reflective Saturable Absorber for Mode-Locked Fiber Laser

**DOI:** 10.3390/nano13101697

**Published:** 2023-05-22

**Authors:** Chi-Cheng Yang, Chih-Hsien Cheng, Ting-Hui Chen, Yung-Hsiang Lin, Jr-Hau He, Din-Ping Tsai, Gong-Ru Lin

**Affiliations:** 1Graduate Institute of Photonics and Optoelectronics and Department of Electrical Engineering, National Taiwan University, Taipei 10617, Taiwan; 2Optical Access Technology Laboratory, Photonic ICT Research Center, Network Research Institute, National Institute of Information and Communications Technology, Koganei 184-8795, Japan; 3Department of Materials Science and Engineering, City University of Hong Kong, Kowloon 999077, Hong Kong, China; 4Department of Electronic and Information Engineering, Hong Kong Polytechnic University, Kowloon 999077, Hong Kong, China

**Keywords:** Ge Saturable Absorber, Ge nanoparticle, saturable absorption, ultrafast fiber laser, passive mode-locking

## Abstract

Amorphous-Ge (α-Ge) or free-standing nanoparticles (NPs) synthesized via hydrogen-free plasma-enhanced chemical vapor deposition (PECVD) were applied as transmissive or reflective saturable absorbers, respectively, for starting up passively mode-locked erbium-doped fiber lasers (EDFLs). Under a threshold pumping power of 41 mW for mode-locking the EDFL, the transmissive α-Ge film could serve as a saturable absorber with a modulation depth of 52–58%, self-starting EDFL pulsation with a pulsewidth of approximately 700 fs. Under a high power of 155 mW, the pulsewidth of the EDFL mode-locked by the 15 s-grown α-Ge was suppressed to 290 fs, with a corresponding spectral linewidth of 8.95 nm due to the soliton compression induced by intra-cavity self-phase modulation. The Ge-NP-on-Au (Ge-NP/Au) films could also serve as a reflective-type saturable absorber to passively mode-lock the EDFL with a broadened pulsewidth of 3.7–3.9 ps under a high-gain operation with 250 mW pumping power. The reflection-type Ge-NP/Au film was an imperfect mode-locker, owing to their strong surface-scattered deflection in the near-infrared wavelength region. From the abovementioned results, both ultra-thin α-Ge film and free-standing Ge NP exhibit potential as transmissive and reflective saturable absorbers, respectively, for ultrafast fiber lasers.

## 1. Introduction

Typical nano-scaled carbon materials with sp^2^ carbon bonds, such as fullerenes, nanotubes, graphene, and graphite, which can convert zero-dimensional (0D) to three-dimensional (3D) structures, have unique properties that support their use in different applications. In the early years of research, one-dimensional (1D) carbon nanotubes transferred from graphite were regarded as saturable absorbers for mode-locked fiber lasers [1]. Thus, two-dimensional (2D) graphene was not successfully isolated from 3D graphite until 2004. Graphene has been extensively researched in recent years owing to its potential applications in electronics, optoelectronics, and photonics. Indeed, 2D materials with excellent performance exhibit new characteristics and are used in various applications, such as electronic contacts, high-performance sensors, and inert coatings [2]. Many 2D materials have been successively synthesized since the development of graphene via either exfoliation or deposition, pioneering a new era of 2D material-based linear and nonlinear photonics. In 2004, Novoselov et al. successfully separated monolayer graphene from graphite [3]. They showed that electrons could fulfill the future development requirements of photocatalysts, sensors, and transparent electrodes by acting as massless Dirac fermions [4,5,6,7,8,9,10,11,12]. Subsequently, several bulk materials (MoS_2_, Bi_2_Te_3_, etc.) capable of exfoliating into 2D matrices bonded via weak van der Waals have attracted research interest. For instance, MoS_2_ exhibits different bandgap energies when transferred from bulk form to 2D sheet form. A bulk MoS_2_ has an indirect bandgap energy of 1.29 eV, whereas a 2D sheet exhibits a direct bandgap of 1.8 eV, facilitating its use in high-mobility transistors [13,14]. Germanene is a 2D material that has recently attracted increasing attention [15,16,17,18]. Similar to graphene, germanene has Dirac fermions and a tunable bandgap. It also exhibits the quantum Hall effect, supporting its use in the electronic and photonic realms [19,20,21]. Although freestanding isolated germanene has yet to be massively produced, Li et al. preliminarily demonstrated that the germanene could be reduced on a Pt (111) surface in 2014 [22]. Dávila et al. further reported that the direct deposition of the germanium atoms on the Au (111) surface could lead to the successful synthesis of few-layered germanene [23]. The formation of a hydrogenated multi-layered germanene from CaGe_2_ was another major breakthrough.

In contrast to 2D materials, semiconductor nanoparticles (NPs), nanopyramids, and nanopillars have recently attracted considerable attention for their functional structures. Many fascinating investigations using NP materials [24] with quantum confinement effects in new scientific and technological applications have been conducted. Instead of bulk Si and Ge, their NPs exhibit efficient luminescence owing to their quasi-direct bandgap and enhanced quantum emission efficiency [25,26,27]. The size-dependent optical characteristics of Si and Ge NPs initiated the development of new branches of photonic devices. Several candidate materials have been proposed for application in communication, frequency comb generation, material modification, and biomedical imaging [28]. Recently, the most intriguing demonstration was a nonlinear saturable absorber for starting up the passive mode-locking fiber lase with these new-era materials. To date, many mechanisms for initiating the passive mode-locking of lasers, including nonlinear polarization rotation (NPR) [29], saturable absorption (SA) [30], and nonlinear optical loop mirror (NOLM) [31], have been explored. Passive mode-locking generated by saturable absorption is currently the main approach for successfully initiating and stabilizing ultrashort laser pulses. In principle, the saturable absorber is a material with intensity-dependent absorption. When the high-intensity pulse was injected into saturable absorption to decrease the material loss of the saturable absorber, redundant light passed through the saturable absorber. Then, the light passed through the gain medium to enhance the optical intensity. Therefore, the laser light oscillated in the laser cavity by following the abovementioned process. The laser intensity was accumulated to induce a mod-locked fiber laser pulse. Various saturable absorbers have been used to start pulsed lasers with different structures and matrices, ranging from bulk/layered semiconductors to 2D topological insulators [32]. Group IV semiconductor saturable absorbers have seldom been investigated for use in mode-lock fiber lasers for femtosecond pulse generation. One of the few to investigate them, Grawert et al. preliminarily fabricated germanium-based saturable Bragg reflectors (SBRs) for mode-locking an erbium-ytterbium:glass laser with a 220-fs pulsewidth at 1550 nm [33]. Recently, 2D semiconductor and topological insulator materials have been used to enable saturable absorption in laser cavities with broadband tunable nonlinear absorption coefficient and ultrafast relaxation time. Among the few reports on mode-locking fiber lasers starting with NP saturable absorbers [34,35], the use of Ge NPs with near-infrared absorption [36] in enabling nonlinear saturable absorption has yet to be explored [37]. Compared to our previous works [37,38], we proposed a similar metal catalyst method for graphene generation [39] to fabricate the Ge-NP-on-Au (Ge-NP/Au) film as the saturable absorber. In this work, the Ge-NP/Au film was easily combined with the cooling system to extend the lifetime of the Ge NPs and maintain mode-locking performance during long-term operation in the future. Similar to 2D materials, such as silicene and germanene, in the post-graphene era [40,41,42], this work provided the possibility of germanene generation for the mode-locked laser in the future.

In this study, a catalyst-assisted self-assembly method was used to synthesize amorphous Ge (α-Ge) on the SiO_2_/Si substrate and free-standing Ge NPs on ultrathin gold (Au) film via hydrogen-free and low-temperature plasma-enhanced chemical vapor deposition (PECVD) at threshold plasma power. The α-Ge and Ge-NP/Au films were adhered onto the end-face of the transmissive and reflective patchcord conductors, respectively, to act as saturable absorbers in a passively mode-lock locked Er-dopes fiber laser (EDFL) system. The Ge-NPs self-assembled on an Au-coated SiO_2_/Si substrate were electrochemically exfoliated via a buffered oxide etching (BOE) solution lift-off process. An Au-Ge phase diagram was used to determine the synthesis parameters for the proper deposition of Ge NPs on Au. Subsequent analyses using scanning electron microscopy (SEM) and atomic force microscopy (AFM) were performed to confirm the size and surface roughness of the free-standing Ge-NP/Au film. A nonlinear optical absorption analysis determined via power-dependent transmission was also performed to determine the saturable absorption of the Ge-NPs. Finally, the performance of the passively mode-locked EDFL with the transmissive α-Ge- and reflective Ge-NPs-based saturable absorber was characterized to realize the ultimate pulsation response after the mode-locking operation. 

## 2. Materials and Methods

### 2.1. Fabrications of α-Ge and Ge-NP/Au Films

The synthesis of α-Ge film and free-standing Ge-NP film on Au-coated SiO_2_/Si substrate was performed in a low-temperature PECVD (EASTERN SHARP LTD., Taoyuan, Taiwan) chamber under hydrogen-free and low-plasma power conditions. Argon-diluted germane (GeH_4_/Ar, MIN YANG SPECIAL GAS CO., LTD., Taoyuan, Taiwan) was used as a precursor in the PECVD system to synthesize the Ge-NP/Au and α-Ge films. The reactant gases were GeH_4_ and Ar, with flows of 0.3 and 2.7 sccm, respectively. The chamber pressure, substrate temperature, and plasma power were 0.3 torrs, 350 °C, and 50 W, respectively. The parameters for the synthesis of the α-Ge and Ge-NP/Au films on SiO_2_/Si substrate in the PECVD system are summarized in Table 1.

A schematic of the exfoliation of the α-Ge and free-standing Ge-NP/Au films is shown in Figure 1. In the initial stage, the Ge-NP/Au and α-Ge films deposited on the SiO_2_/Si substrate were immersed in a buffer oxide etching (BOE, ECHO CHEMICAL CO., Miaoli LTD., Taiwan) solution to etch the SiO_2_ layer and exfoliate both films. After lift-off exfoliation, the Ge-NP/Au and α-Ge films were rinsed in deionized (DI) water, which was gradually injected to dilute the BOE solution and thereby preserve aspects of the Ge-NP/Au and α-Ge films for transfer to the end-face of the patchcord connectors.

### 2.2. Nonlinear Transmittance Analysis of the α-Ge and Ge-NP/Au Films

The experimental setup for analyzing the nonlinear optical transmittance of the α-Ge and Ge-NP/Au films is depicted in Figure 2. A pulsed fiber laser with a pulsewidth of 700 fs and a central wavelength of 1560 nm was employed as the pumping source. To reinforce the peak intensity of the incident optical pulse, an erbium-doped fiber amplifier (EDFA, SDO COMMUNICATIONS CORP., Fremont, CA, USA) with a power gain of 21 dB was used. An optical attenuator was used to continuously adjust the power of the incident optical pulse, ranging from −30 to 0 dB. An output coupler divided the incident light into two beams, where 90% of the optical power that passed through the α-Ge or Ge-NP films was collected by a power meter (P_2_), and the remaining 10% was directly sent to another power meter to serve as a reference (P_1_). Eventually, the exact transmittance of the α-Ge or Ge-NP/Au films was calculated using the ratio P_2_/P_1_.

### 2.3. Setup of Transmissive- and Reflective-Type Mode-Locked EDFL Systems

A transmission-type passively mode-locked EDFL system was constructed, as shown in Figure 3a. A 980-nm laser diode (forward pump, Research Lab Source Inc., Bozeman, MT, Canada) and a 1480-nm laser diode (backward pump, Research Lab Source Inc., Bozeman, MT, Canada) were used to pump the gain medium of 2-m erbium-doped fiber (EDF, nLIGHT Liekki Er80-8/125, nLight Inc., Camas, WA, USA). The EDF and single-mode fiber (SMF, Corning Inc., Corning, NY, USA) had dispersion coefficients (*β*_2_) of −22.6 and −20 ps^2^/km, respectively. The polarization controller adjusted the circulating polarization in the EDFL cavity. An optical isolator was inserted to define the direction of light circulation. The 1 × 2 coupler provided 5% output coupling and a 95% feedback ratio. In addition to the EDF, the 4.8-m SMF formed the residual EDFL cavity. The total group delay dispersion (GDD) of the EDFL system was −0.1412 ps^2^. In contrast, Figure 3b compares a reflection-type passively mode-locked EDFL system with a transmission-type system. A patchcord connector with an end-face coated with a 30-nm-thick Au film via a thermal e-gun (reflection > 97%) was employed as a reflector. The Ge-NP/Au film was transferred to the end-face of the Au-coated patchcord to enhance reflectance and reduce cavity loss. Subsequently, the patchcord was linked to the EDFL cavity via a connection with port 2 of the circulator, while the other ports (ports 1 and 3) were connected to the EDFL system. The insets of Figure 3a,b show images of the SMF patchcord end-face with the transferred α-Ge film and Ge-NP/Au films, respectively.

## 3. Results and Discussion

### 3.1. Material and Nonlinear Optical Properties of the Ge-NP/Au and α-Ge Films

Figure 4a shows the assessed Au-Ge phase diagram employed for realizing the successful deposition of free-standing Ge-NPs on the Au-coated SiO_2_/Si substrate [43]. To accurately control the adsorption and reduction mechanisms of Ge atoms in the Au matrix for growing Ge-NPs on an ultrathin Au film-based catalyst, the Ge-NPs could be self-assembled on the Au membrane in a low-temperature and hydrogen-free environment. The Ge atoms were initially dissolved in the Au host during synthesis. The Au-Ge alloy momentarily existed above an appropriate temperature. By gradually cooling the ultrathin Au membrane, the solubility of the Ge atoms in the Au host gradually decreased, resulting in the dissolved Ge atoms gradually desorbing from the Au host and precipitating on the surface of the Au film. 

Figure 4a shows the zoomed-in version of the Au-Ge phase diagram, where the at% of Ge varies from 0% to 5%, and the temperature ranges from 200 °C to 500 °C. When the Au substrate was heated beyond 250 °C, the Ge atoms began to dissolve into the Au host, establishing a solid solution. Figure 4b shows a schematic diagram of the Ge atoms dissolved in the Au host. The dissolved Ge atoms were randomly located in the interstices between the Au atoms, inducing the subsequent precipitation of Ge-NPs, as shown in Figure 4b. When the Au substrate was cooled to room temperature, the solubility of the Ge atoms in the Au matrix decreased, causing Ge atoms to precipitate on Au films. The low-temperature Ge dissolution precisely controlled the quantity of Ge film and determined the size of Ge-NP during the slow cooling process. Moreover, radio-frequency (RF) plasma was used to enhance the decomposition of GeH_4_ in a hydrogen-free environment and to guarantee the quantity of dissolved Ge atoms.

The top-view AFM images of the nanostructured α-Ge film grown on a silicate coverslip glass plate under different deposition durations are shown in Figure 5. After scanning the cross-section profile of the α-Ge film from X_1_ to X_2_ horizontally, the mapped morphological contours of the α-Ge films versus the *X*-axis position were analyzed and compared among one another. The surface roughness factor (*M*) of the α-Ge film is defined by the root-mean-square method, as described by
(1)$$M=\sqrt{\frac{\sum_{i=1}^{n}X_{i}{}^{2}}{n}}=\sqrt{\frac{X_{1}{}^{2}+X_{2}{}^{2}+X_{3}{}^{2}+\ldots +X_{n}{}^{2}}{n}}$$
where *X* denotes the deviation of the α-Ge thickness from the reference plane, and *n* is the sample position point along the horizontal direction. Based on this calculation, the root-mean-square surface roughness of the α-Ge films grown under different deposition times of 15, 30, and 60 s was evaluated as 1.147, 1.629, and 2.298 nm, respectively. The α-Ge film was synthesized with a longer deposition time, exhibiting a flatter surface with a larger nano-rod size of the roughened α-Ge array.

In general, the Mie scattering effect could be neglected in the nanostructured α-Ge surface roughness, as its surface roughness was well below λ/500, and only the Rayleigh scattering would predominate the surface scattering of the EDFL beam caused by the nanostructured α-Ge rods with their size of much smaller than the EDFL wavelength. The optical intensity passing through the scattered α-Ge film surface was proportional to the product of D^6^/λ^4^ with D and λ, respectively, denoting the average rod diameter of the deposited α-Ge array. Although the 60-s-deposited α-Ge film exhibited a Rayleigh scattering phenomenon 32 times stronger than the 15-s deposited α-Ge film, size/wavelength dependence pointed out that the Rayleigh scattering effect was also minimized in all nanostructured α-Ge films in this work. The scattering loss of the nanostructured α-Ge film could be ignored to avoid the additional linear transmission loss caused by the roughened α-Ge array, further assisting the start-up of the self-amplitude modulation procedure, as well as the passive mode-locking process in the EDFL. For a saturable absorber, the modulation depth of the nano-roughened α-Ge array could be described as q_mode-locking_ = q_o_ + q_non_{1 + [I_in_(1 − R)(1 − e^−ΣαL^)]/I_sat_}^−1^ ≈ q_o_ + q_non_{1 − [I_in_(1 − R)(1 − e^−ΣαL^)]/I_sat_} ≡ q_o_ + q_non_ [1 − δ_SAM_I_in_], where α_o_, α_non_, I_in_, R, Σα, L, I_sat_, and δ_SAM_ denote the linear modulation depth, nonlinear modulation depth, incident laser intensity, surface reflectance, total (including absorption + scattering) loss, interaction length, saturation intensity, and self-amplitude modulation (SAM), respectively, of the nano-roughened α-Ge array. Obviously, both the reduction of the surface reflectance and the total loss of the saturable absorber were effective to the minimization of the δ_SAM_, as well as the enhancement of the q_mode-locking_, for improving the mode-locking performance.

The AFM images of the Ge-NP/Au films under different deposition times are shown in Figure 6. Self-assembly Ge islands scattered on the Au film were observed in these images. The area density of the Ge-NPs significantly decreased with the reduction in the deposition time. The root-mean-square heights of the Ge-NP/Au film under deposition durations of 15, 30, and 60 s were 30.35, 35.40, and 56.86 nm, respectively, revealing that the lower deposition time induced shorter Ge islands due to the fewer Ge atoms dissolved in the Au film. The size of the Ge-NPs was characterized by scanning electron microscopy (SEM), as shown in Figure 6. Under deposition durations of 15, 30, and 60 s, the corresponding sizes of Ge-NPs were 143, 321, and 393 nm, respectively. As a result, the vertical or horizontal aspect size of the Ge-NP/Au film showed an increasing trend under increased deposition duration, exhibiting the aspect/wavelength ratios of 1/50~1/25 (in longitudinal cross-section) and 1/10~1/5 (in transverse cross-section). Therefore, the EDFL illumination would induce the Mie scattering effect in the transverse direction (perpendicular to the Ge NP surface) when the Rayleigh scattering effect broke down at the aspect/wavelength ratio of around 1/10. In contrast to the dependence of D^6^/l^4^ for Rayleigh scattering, Mie scattering exhibited a relationship of 1/l^4^D^2^ to be significantly larger than the former one. Therefore, the Mie scattering of the Ge-NP/Au film inevitably induced a larger scattering loss, definitely degrading the SAM coefficient to result in an imperfect mode-locking result compared to the nano-roughened α-Ge array synthesized on a coverslip glass plate.

In this work, Raman scattering analysis using a He-Ne laser (Newport Corporation, Irvine, CA, USA) with a wavelength of 633 nm and a pump intensity of 0.8 mW/mm^2^ was utilized to observe the structural bonds in the Ge saturable absorber. The micro-Raman scattering spectra of the Ge-NPs on the Au membrane and α-Ge film on glass are shown in Figure 7. The Raman peak located at 300 cm^−1^ was due to the Ge-Ge vibration mode of crystalline Ge [44]. Regarding the Ge-NPs, the peak positions of the Ge-Ge vibration mode for all samples slightly deviated from 300 cm^−1^ [44], resulting in a blueshift in its wavenumber from 296.9 to 298.1 cm^−1^ with an increase in the deposition time from 15 to 60 s. The blueshift of the Ge-Ge vibration mode was mainly attributed to the decreased size of the Ge-NPs. In principle, a phonon confinement model for the first-order Raman scattering spectrum was used to simulate the scattering peak intensity of the Ge-Ge vibration mode, as described in [44,45].
(2)$$I(\omega )=\oint_{0\to 1}e^{\frac{-q^{2}d^{2}}{4a^{2}}}\times \frac{4\pi q^{2}}{{\left\{[\omega -\omega (q)]\right\}}^{2}+\langle \frac{\Gamma_{0}{}^{2}}{2}\rangle }dq$$
where *a* is the lattice constant of Ge, *d* is the size of Ge-NP; *ω*(*q*) is the transverse optical (TO) phonon dispersion frequency related to *q*, and Γ_0_ is the natural linewidth of the TO phonon in crystalline Ge. In addition, *q* is expressed in terms of 2*π*/*a*. After simplification, the wavenumber shift of the Ge-Ge vibration mode is associated with the size of the Ge-NP as follows [44]:(3)$$\Delta \omega =\omega (q)-\omega_{o}=-A{\left(\frac{a}{d}\right)}^{\gamma },$$
where *A* and *γ* are constants. Consequently, a smaller Ge-NP can lead to a larger wavenumber shift. In contrast, Figure 7 shows the Raman scattering spectrum of α-Ge on a glass substrate. The peak positions among all samples located at 170 and 270 cm^−1^ were related to the Ge-Ge transverse acoustic (TA) and TO modes, respectively [46]. In addition, the strain in the Ge-Ge bonds also induced the Raman peak shift. From previous works [47,48], the tensile strain in the Ge-Ge bonds usually caused the redshift of the Raman peak. In this work, the Raman peak was blue shifted when the deposition time increased from 15 to 60 s to indicate that the tensile strain became weak owing to the phase transition of the Ge-Ge bond from amorphous and crystalline and the thermal expansion difference between Ge and Au [49].

### 3.2. Nonlinear Transmissive Analysis of Transmissive-Type α-Ge and Reflective-Type Ge-NP/Au Films

To facilitate comparison among the samples, the transmittances of the α-Ge films synthesized under different deposition times under a lower incident power were measured at ~0.9. Therefore, the nonlinear optical properties of the three samples could be compared without considering the offset, owing to their linear transmittances. Owing to the high linear transmittance, the linear absorption loss of the α-Ge film remained relatively small, effectively scaling down the mode-locking threshold of the EDFL. For comparison, Figure 8a shows the nonlinear transmittance and normalized absorbance of the α-Ge films synthesized at different deposition times. Under pumping by a pulsed fiber laser with a central wavelength of 1560 nm and a pulsewidth of 700 fs, the transmittance of the α-Ge film grown under a deposition time of 60 s increased from 0.936 to 0.970 in increments of 0.034 (3.6%). Regarding the α-Ge film grown under a deposition time of 30 s, the transmittance increased from 0.942 to 0.978 in increments of 0.036 (3.8%). When the deposition time decreased to 15s, the transmittance of the α-Ge film increased from 0.944 to 0.982 in increments of 0.038 (4.0%).

To fairly compare the nonlinear optical properties of the α-Ge films, the response of nonlinear transmittance vs. peak power was curve-fitted using the equation T = exp{−q_lin_ − [q_non_/(1 + P_in_/P_sat_)]}, where q_lin_, q_non_, and P_sat_ denote the linear absorbance, nonlinear absorbance, and saturation power, respectively. For example, the parameters q_lin_, q_non_, and P_sat_ of the α-Ge film grown under a deposition time of 60 s were 1.1 × 10^−2^, 5.45 × 10^−2^, and 61.25 W, respectively. The modulation depth calculated by the normalized absorbance was 52%. When the deposition time was reduced to 15 s, the parameters q_lin_, q_non_, and P_sat_ of the α-Ge film varied to 2.7 × 10^−2^, 5.43 × 10^−2^, and 60.73 W, respectively. The corresponding modulation depth increased from 52% to 58%. An analysis of the parameters related to passive mode-locking in Table 2 revealed that the most significant factor for self-starting mode-locking at the initial stage was the SAM coefficient (γ), defined as q_non_/P_sat_. The coefficient not only determined the mode-locking force but also impacted the shaping of the pulsewidth, as shown by τ = (2D_g_ × I_peak_/γ)^1/2^, where D_g_ denotes the gain dispersion. A saturable absorber with a large SAM coefficient and high modulation depth is an excellent candidate for modulating EDFL systems. Based on the aforementioned discussion, an α-Ge film with a deposition time as short as 15 s is more suitable than other samples for serving as a transmissive-type saturable absorber in a passively mode-locked EDFL system.

An Au-coated patchcord was employed to host the transferred Ge-NP/Au film and measure its saturable absorbance. Another patchcord was used to connect the free-standing Ge-NP/Au plated patchcord. The connected patchcords were linked to a circulator (port 2). The EDFL output was inserted into port 1 of the circulator. The light reflected by the Ge-NP/Au film at port 2 was collected at port 3. After subtracting the loss caused by the circulator from the reflectance, the nonlinear transmittance of the Ge-NP/Au film was obtained from the analysis, as shown in Figure 8b. Table 3 shows the nonlinear transmittance, normalized absorbance, and other characteristic parameters of the Ge-NP/Au film grown under different deposition times. When the incident intensity passing through the Ge-NP/Au film grown under a deposition time of 60 s increased, the reflectance increased from 0.417 to 0.431 in increments of 0.014 (9.5%). When the deposition time was shortened to 30s, the reflectance of the Ge-NP/Au film was increased from 0.430 to 0.445 in increments of 0.015 (5.5%). Further shortening the deposition time to 15s increased the reflectance of the Ge-NP/Au film from 0.528 to 0.547 in increments of 0.019 (4.3%). The intracavity light experienced by the reflection-type saturable absorber was twice that of the Ge-NP saturable absorber, showing similar increments in transmittance under nonlinear saturable absorption. However, the linear transmittance was significantly attenuated owing to the round-trip propagation through the Ge-NP/Au film twice.

### 3.3. Mode-Locking Performances of the Transmissive-Type α-Ge and Reflective-Type Ge-NP Saturable Absorber-Enabled Ge-Based EDFL System

Figure 9 shows the optical spectra and autocorrelation trace of the passively mode-locked output at near-threshold pumping conditions obtained from different α-Ge films after inserting the α-Ge or Ge-NP film for mode-locking the EDFL. The threshold pumping power for the 980-nm LD was approximately 41 mW. Under these conditions, the output pulse intensities were too small to be detected by the autocorrelator. To overcome this problem, an EDFA was used to linearly amplify the output peak intensity. Consequently, the α-Ge saturable absorbers synthesized under different deposition times of 15, 30, and 60 s individually mode-locked the EDFL with output pulsewidths of 700, 702, and 707 fs and corresponding spectral linewidths of 6.11, 5.88, and 5.60 nm, respectively. At the mode-locking threshold, the SAM mechanism dominated the EDFL pulsation. Regarding the α-Ge film grown under different deposition times, the EDFL produced an almost identical pulsewidth.

The pulse trains of the passively mode-locked EDFL with α-Ge saturable absorbers shown in Figure 9a revealed that the carrier amplitude jitters (CAJs, defined as σ/I_a_, where σ is the standard deviation of the pulse intensity, and I_a_ is the average pulse intensity) [32] were 5.72%, 4.82%, and 4.19% when using α-Ge film grown under deposition times of 15, 30, and 60 s, respectively. When operating the EDFL at the mode-locking threshold, the peak power of the pulse train fluctuated severely in the time domain with a high CAJ, revealing unstable pulsation, even under the mode-locked condition. When the EDFL was mode-locked under the high-gain condition with pumping power as high as 155 mW, the α-Ge saturable absorbers grown under different deposition times of 60 s could enable the passive mode-locking of the EDFL at a central wavelength of 1575 nm with a pulsewidth below 300 fs and a corresponding spectral linewidth broader than 8.58 nm. When the deposition time of the α-Ge film was lowered from 30 to 15 s, the mode-locked pulsewidth was further decreased from 294 to 290 fs, with corresponding spectral linewidth increasing from 8.83 to 8.95 nm owing to the larger modulation depth for the α-Ge film grown under a shorter period. Under high-pumping conditions, the time-bandwidth products (TBPs) of the passively mode-locked EDFLs were approximately 0.315, almost characteristic of transform-limited conditions. Concurrently, the relative CAJs of the passively mode-locked EDFLs improved to less than 1%, indicating that the stabilized mode locking of the EDFL could be achieved with the relatively flattened envelope of the mode-locked pulse train in the time domain.

In particular, the passively mode-locked EDFL spectra shown in the middle column of Figure 9 also confirmed that mode-locking had entered the soliton regime, which was attributed to the Kelly sidebands (observed on the shoulder of optical spectra) induced by the periodic perturbation of linear anomalous group delay dispersion (GDD) and nonlinear self-phase modulation (SPM). This periodic perturbation resulted in a nonlinear index grating that strengthened the dispersive waves at specific wavelengths after round-trip circulation in the EDFL cavity. When the dispersive wave was phase-matched with the soliton pulse, sharpened parasitic spectral sidebands appeared on the pedestals of the mode-locked spectrum [50,51]. After self-starting the EDFL pulse through the SAM mechanism, soliton compression under a high-gain regime was induced by compensating for the effects of GDD and SPM, which reshaped the pulse envelope. Additionally, the α-Ge saturable absorber plays an important role in stabilizing the EDFL pulse at the initial pulse-forming stage [52,53].

The transmissive-type and reflective-type Ge-NP saturable absorber-enabled passively mode-locked fiber lasers are compared in Figure 10, which shows the mode-locking performance enabled by the reflection-type system with the Ge-NP/Au film. The cavity loss exhibited by the reflection-type saturable absorber system, which mainly resulted from the circulator and scattering by Ge-NPs, was larger than that exhibited by the transmission-type saturable absorber system. Owing to this large cavity loss, the pumping power in the reflection-type system inevitably increased (to 249.6 mW) to compensate for the cavity and provide an equivalent cavity gain to that of the transmission-type system; another EDFA was employed to amplify the insufficient peak power of the mode-locked EDFL pulses. Regarding the Ge-NP/Au saturable absorber film grown under different deposition times of 30 and 60 s, weak and broadened EDFL pulses with pulsewidths of 3800 and 3950 fs, respectively, were observed owing to the small modulation depth of the SAM at the initial stage. When the Ge-NP deposition time was shortened to 15 s, the scattering loss of the thinner Ge-NP film effectively reduced, resulting in an EDFL pulsewidth of 3700 fs. The CAJs of the EDFL passively mode-locked by the Ge-NPs under deposition times of 15, 30, and 60 s were 1.45%, 1.78%, and 3.09%, respectively. Undoubtedly, the large cavity loss in the reflection-type system not only increased the mode-locking threshold but also led to the generation of a broadened EDFL pulse under the SAM regime. Even when the pumping power was increased to 250 mW (the highest pumping power limited by the 980-nm LD), soliton mode-locking was hardly induced, further shortening the EDFL pulse. Additionally, the small area density of the Ge-NPs synthesized on the Au film led to an insufficiently large SAM coefficient, which failed to provide a large mode-locking force for sharpening the EDFL pulse after mode-locking.

## 4. Conclusions

The α-Ge and Ge-NPs synthesized via hydrogen-free PECVD were used as transmissive or reflective saturable absorbers, respectively, to create passively mode-locked EDFLs. When the deposition time decreased from 60 to 15 s, the root-mean-square heights of the Ge-NPs decreased from 56.86 nm to 30.35 nm, resulting in the Raman scattering signal of the Ge-NPs/Au, deviating from the signal of the crystalline Ge at 300 cm^−1^, which ranged from 3.1 to 1.9 cm^−1^. This indicated that the suppression of Ge-NP increased with decreasing deposition time. The linear transmittance of α-Ge slightly increased from 92.4% to 95.4% as the deposition time decreased from 60 to 15 s. Decreasing the deposition time slightly increased the transmission variation from 3.4% to 3.6%, which in turn increased the modulation depth from 52% to 58%. Moreover, the modulation depth for the reflection-type Ge-NP saturable absorber also increased from 4.3% to 5.5%. Regarding the transmissive-type EDFL, the α-Ge grown under a deposition time of 15–60 s could self-start the EDFL pulsation and produce a pulsewidth of 700–707 fs. Under the high-gain pumping operation, the pulsewidth of EDFL increased from 290 to 300 fs, with the corresponding spectral linewidth decreasing from 8.95 to 8.58 nm because of the soliton compression by the SPM effect, resulting in the pulsation being reshaped. The reflective-type Ge-NP saturable absorber grown under the deposition time of 15–60 s provided the narrow-band optical spectrum with an increase in the pulsewidth from 3700 to 3950 fs under the high-gain operation of 249.6 mW, indicating that incompletely passive mode-locking was preferable for this EDFL. The Ge-NPs exhibited a large modulation depth, delivering short pulsations for passive mode-locking. Although the reflective-type EDFL passively mode-locked by the Ge-NPs exhibited imperfect mode-locking, the Ge-NPs showed potential as mode-lockers for ultrafast fiber lasers.

## Figures and Tables

**Figure 1 nanomaterials-13-01697-f001:**
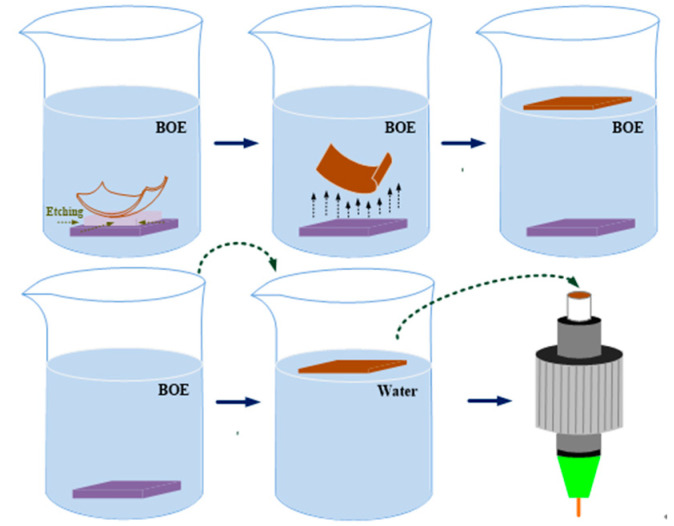
Process of exfoliating and adhering films on the end-face of a single-mode fiber (SMF).

**Figure 2 nanomaterials-13-01697-f002:**
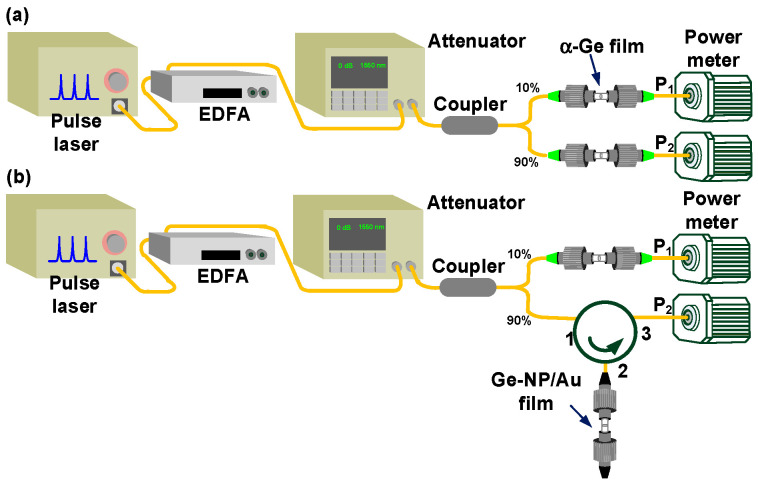
Schematic diagram for measuring the nonlinear transmittance of the α-Ge and Ge-NP/Au films which serve as the (**a**) transmission-type and (**b**) reflection-type saturable absorbers, respectively, in the mode-locked EDFL system.

**Figure 3 nanomaterials-13-01697-f003:**
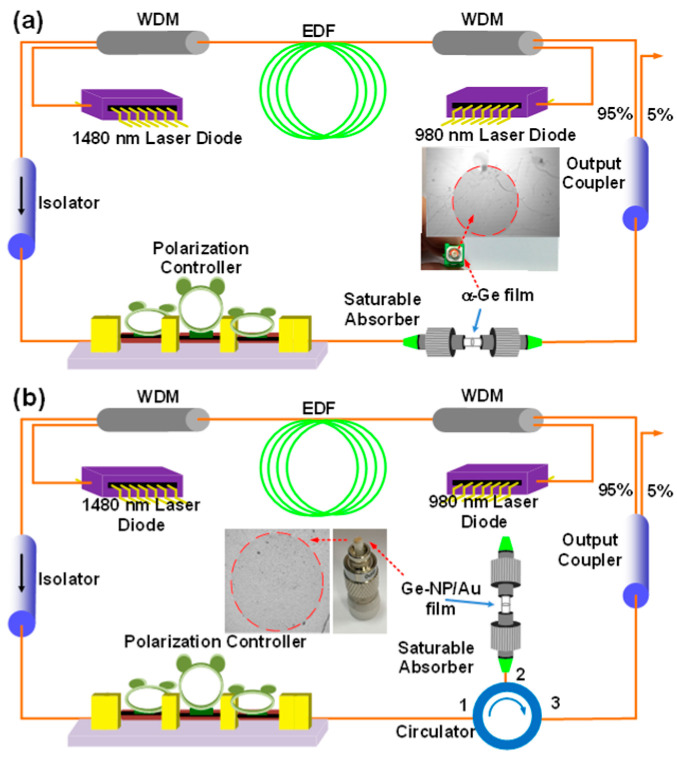
Schematic diagram of the (**a**) transmission-type and (**b**) reflection-type mode-locked EDFL system, with the Ge film serving as a saturable absorber.

**Figure 4 nanomaterials-13-01697-f004:**
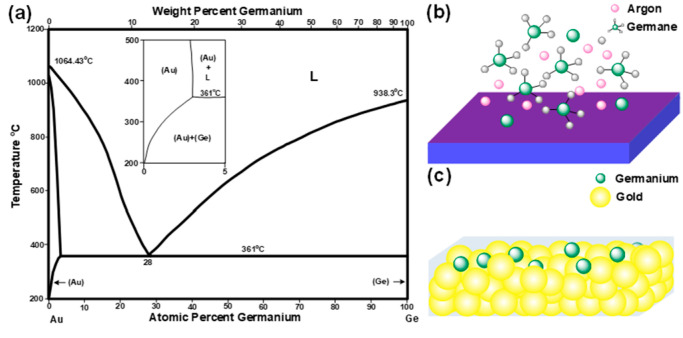
(**a**) Phase diagram of Au-Ge. Schematic diagrams of (**b**) Ge atoms directly deposited on the thermal oxide wafer using a hydrogen-free PECVD system and (**c**) Ge atoms dissolved into the Au host using a hydrogen-free PECVD system.

**Figure 5 nanomaterials-13-01697-f005:**
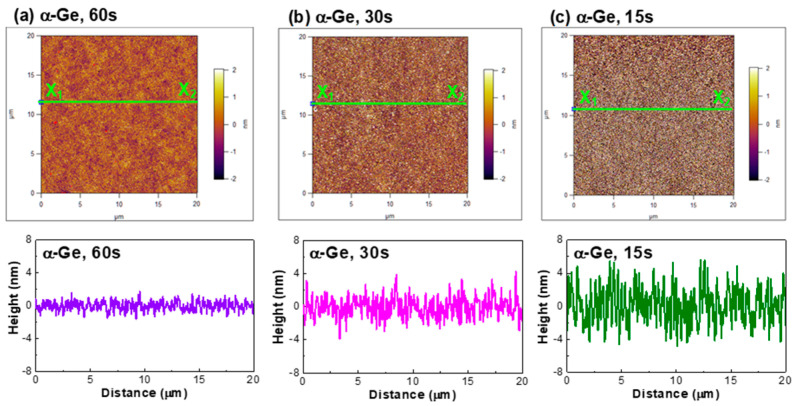
AFM images of the α-Ge film on the SiO_2_/Si substrates under different deposition times: (**a**) 60, (**b**) 30, and (**c**) 15 s.

**Figure 6 nanomaterials-13-01697-f006:**
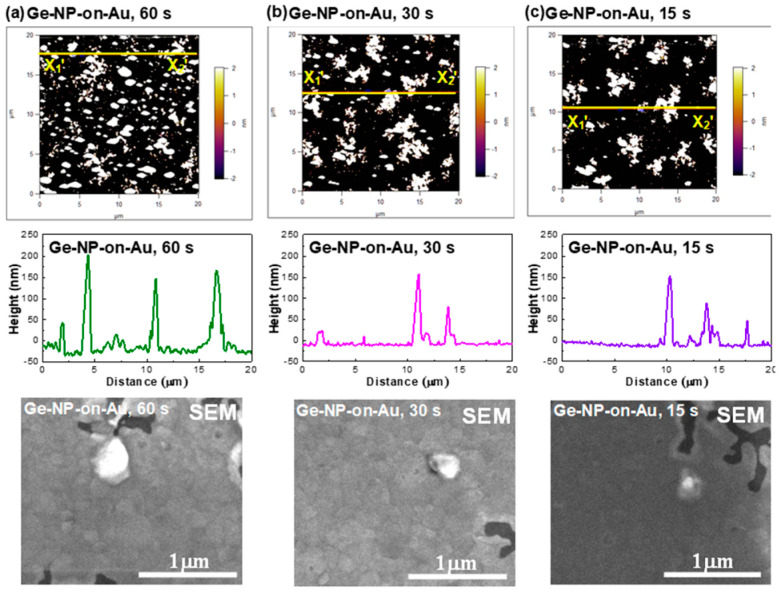
AFM and SEM images of the Ge-NP/Au film under different deposition times: (**a**) 60, (**b**) 30, and (**c**) 15 s.

**Figure 7 nanomaterials-13-01697-f007:**
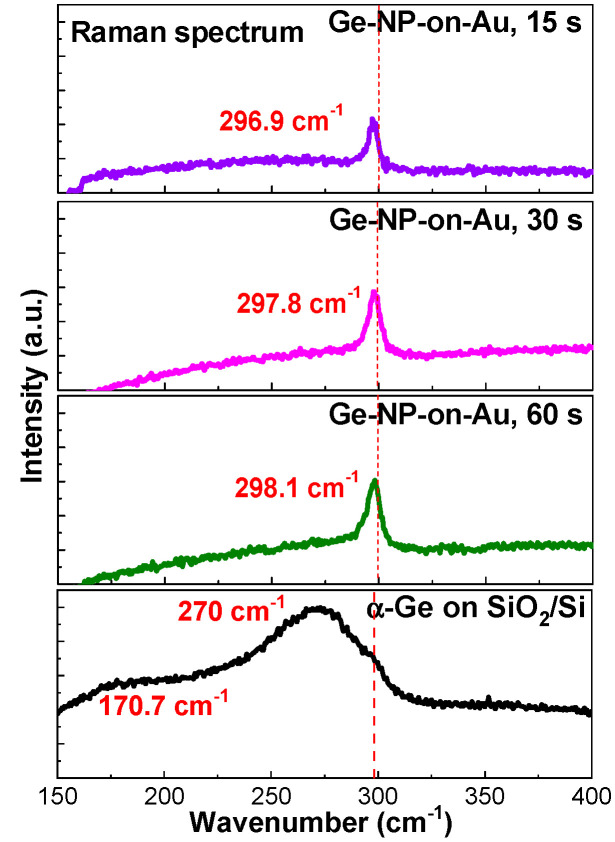
Raman scattering spectra of α-Ge and Ge-NP/Au films under different deposition times.

**Figure 8 nanomaterials-13-01697-f008:**
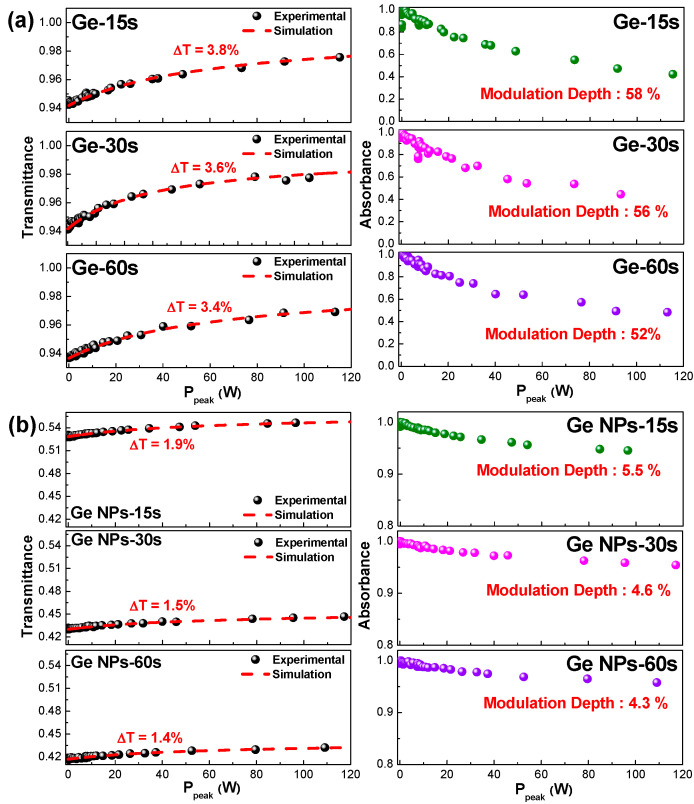
Nonlinear transmittance and normalized absorbance of the (**a**) α-Ge and (**b**) Ge-NP/Au films synthesized at different deposition times.

**Figure 9 nanomaterials-13-01697-f009:**
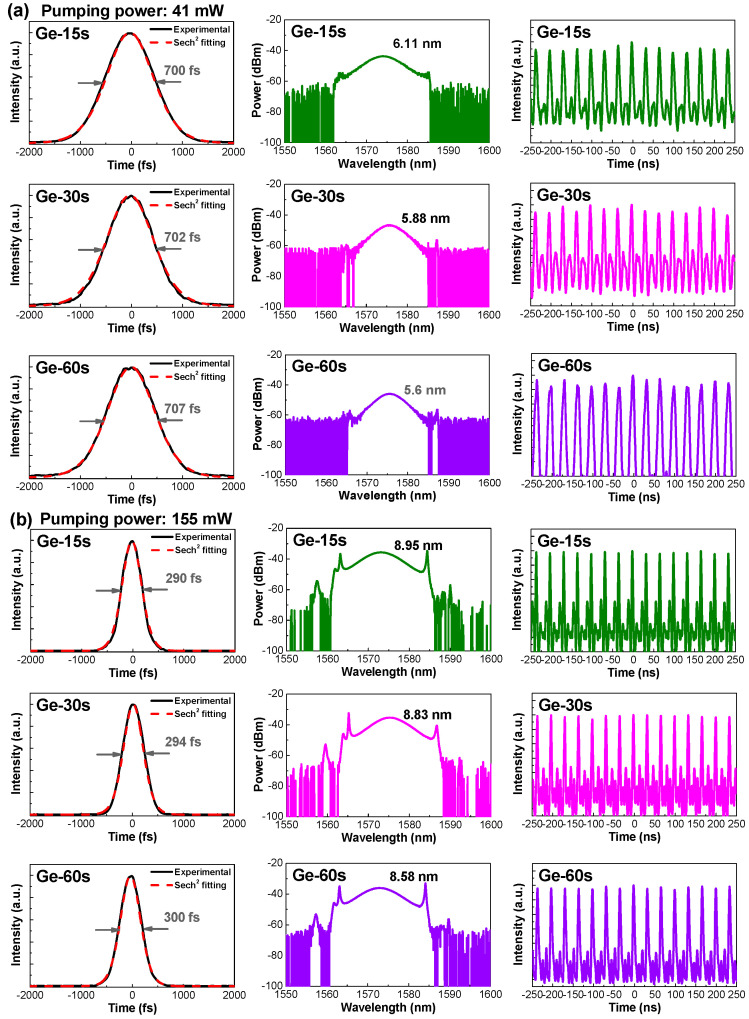
Autocorrelation traces (**left**), optical spectra (**center**), and pulse train (**right**) of the transmission-type passively mode-locked EDFLs at pumping powers of (**a**) 41.16 mW and (**b**) 155.1 mW, with α-Ge saturable absorbers grown under different deposition times.

**Figure 10 nanomaterials-13-01697-f010:**
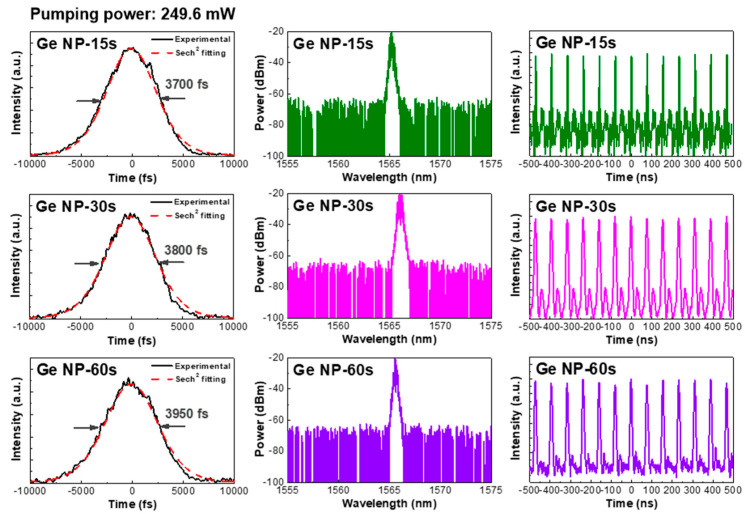
Autocorrelation traces (**left**), optical spectra (**center**), and pulse train (**right**) of the reflection-type saturable absorber-enabled passively mode-locked EDFLs at a pumping power as high as 249.6 mW using the Ge-NP/Au films grown under different deposition times.

**Table 1 nanomaterials-13-01697-t001:** Relative parameter for the synthesis of the α-Ge and Ge-NP/Au films on a SiO_2_/Si substrate in the PECVD system.

Deposition Time (s)	15	30	60
[GeH_4_/Ar] (sccm)	0.3:2.7		
Temperature (°C)	350		
Power (W)	50		

**Table 2 nanomaterials-13-01697-t002:** Nonlinear transmittance parameters of the exfoliated α-Ge films grown at different deposition times.

Deposition Time (s)	15	30	60
Linear absorbance q_lin_	0.011	0.0061	0.0056
Nonlinear absorbance q_non_	0.0545	0.054	0.0543
Saturation power P_sat_ (W)	61.25	60.42	60.73
SAM coefficient	8.90 × 10^−4^	8.93 × 10^−4^	8.94 × 10^−4^
Modulation depth	58%	56%	52%

**Table 3 nanomaterials-13-01697-t003:** Nonlinear transmittance parameters of exfoliated Ge-NP/Au films at different deposition times.

Deposition Time (s)	15	30	60
Linear absorbance q_lin_	0.011	0.0061	0.0056
Nonlinear absorbance q_non_	0.0545	0.054	0.0543
Saturation power P_sat_ (W)	61.25	60.42	60.73
SAM coefficient	8.90 × 10^−4^	8.93 × 10^−4^	8.94 × 10^−4^

## Data Availability

The data presented in this study are available on request from the corresponding author.

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
