# Peer review of "Synthesis of Nano-Structured Ge as Transmissive or Reflective Saturable Absorber for Mode-Locked Fiber Laser"

_nanomaterials, 2023, doi:10.3390/nano13101697_

Round 1

Reviewer 1 Report

-          The authors didn’t discussed results of manuscript with the ones from Opt. Laser Technol. 2021, 136, 106761, please discuss improvements/novelty considering previous articles [Opt. Laser Technol. 2021, 136, 106761] and [Nanomaterials 2022, 12, 1197]. SAM coefficient of 8.90×10-4 was reported in both manuscript nanomaterials-2337016-peer-review-v1 and Opt. Laser Technol. 2021, 136, 106761 for amorphous Ge.

-          The authors need to improve the morphological study by better discussion and interpretation of results

-          Lines 90-92 “Among few reports on the mode-locking fiber laser started with NP saturable absorber [34, 35], the Ge NPs with near-infrared absorption [36] have yet to be explored for enabling the nonlinear saturable absorption to date [37].” Reference 36 presence absorption for 230 – 440 nm interval in UV and beginning of VIS. Please see Reference [Sci. Rep. (2018) 8:4898, DOI:10.1038/s41598-018-23316-3] with amorphous Ge NPs with absorption limit of 0.73 eV, while for Ge NCs the absorption threshold is 1.14 eV.

-          Lines 211-217 related to Ge NPs sizes by AFM, it results that their height is 56.85, 35.40 and 30.35 nm respectively function of deposition time, while their lateral size is 393, 321 and 143 nm respectively. Please write more clearly the morphological characteristics of Ge NPs, are they bulk-like?

-          Specify Raman measurement conditions including laser wavelength.

-          Check line 229 with references 40 and 41, I think they are wrong. Formula (2) is from Reference 39.

-          Line 233: Reference 42 is wrong, it is reference 40

-          Formula 2 is missing 4piq2dq. Please see correct formula in [formula 1 in J Nanopart Res (2013) 15:1981, DOI 10.1007/s11051-013-1981-y], and also related to lines 232-234, the phonon dispersion used in the Lorentz function given by Nilsson and Nelin (1971) for crystalline Ge in [formula 2 in J Nanopart Res (2013) 15:1981, DOI 10.1007/s11051-013-1981-y].

-          By computing formula from in J Nanopart Res (2013) 15:1981, and normalizing the intensity function of diameter it results that the peak position of Ge-Ge maximum red-shifts (to lower energies) compared to bulk Ge (299.8 cm-1) with the decrease of diameter due to phonon confinement in NCs. Red-shift of peak can be produced also by strain. From the manuscript it results that Ge NPs are bulk-like, with heights exceeding 30 nm and lateral size of more than 100 nm, so phonon confinement is excluded, it remains strain factor to be considered and discussed.

Author Response

National Taiwan University

Graduate Institute of Photonics and Optoelectronics (GIPO)

Department of Electrical Engineering

1, Roosevelt Rd. Sec. 4, Taipei 10617, Taiwan R.O.C.

------------------------------------------------------------------------------------------------

Dear Editor:

We would like to submit the revised manuscript (Article Number: nanomaterials-2337016) entitled “Synthesis of Nano-structured Ge as Transmissive or Reflective Saturable Absorber for Mode-locked Fiber Laser” for your consideration of publication in Photonics. This manuscript has been carefully revised according to reviewer’s comments in separated files. Please allow us the opportunity to consider the manuscript for publication. We sincerely appreciate your kind help.

Best regards,

Gong-Ru Lin

=========================================

Y. Z. Hsu Science Chair Professor

NTU Lifetime Distinguished Professor

Distinguished Research Fellow of MOST

FIEEE, FOSA, FSPIE, FIET, FInstP

Society Directors of SPIE

13th President of Taiwan Photonics Society

Associate Dean, College of Electrical Engineering and Computer Science

Graduate Institute of Photonics and Optoelectronics

Department of Electrical Engineering

National Taiwan University

OSA Traveling Lecturer and SPIE Visiting Lecturer

Senior/Associate Editor, IEEE Photonics Journal

Topical Editor, Optics Letters

No. 1, Sec. 4, Roosevelt Rd., Taipei 10617, Taiwan

E-mail: grlin@ntu.edu.tw

Web: http://scholar.google.com/citations?user=QfFMP0cAAAAJ&hl=en

Phone: +886-2-33663700 ext. 6519

Fax: +886-2-33669598

Reviewer 2 Report

Manuscript: “Transmissive amorphous Ge Film or Reflective Ge Nanoparticles on Gold
as Saturable Absorber for Mode-locked Fiber Lasers”

Authors: Chi-Cheng Yang, Chih-Hsien Cheng, Ting-Hui Chen, Yung-Hsiang Lin,
Jr-Hau He, Din-Ping Tsai, Gong-Ru Lin *

It seems that this paper, reporting the synthesis and the investigation (by using several techniques) of amorphous Ge and/or Ge nanoparticles to be used as the end-face transmission/reflection types for “saturable” absorber in the mode-locked erbium doped-fibre laser, could be interesting if some suggestions and comments below reported will be addressed.

First of all, the title is too long and not immediately related of the work performed.

The abstract is completely confused, as well as in other part of the text.
Add some simple sentences to help the reader to understand this mode locked erbium-fibre laser, and why this work can advance the research in the field.
Cite silicene as the father of 2D materials (PRL, 108, 15501, 2012) in the post graphene era, indeed it was before the germanene.

It is very important that the paper will be revised by a native speaker. Disseminated in the text there are errors and mistakes (it means related to wrong translation). I mean, I can understand the sense of the sentences, but they are wrong.

In the present form the manuscript can’t  be published in Nanomaterials.
I hope that the next version will be emended  by that, making possible a correct reading, without  finding to understand what the authors would have meant.

Author Response

(The authors gave the same response as above.)

Round 2

Reviewer 1 Report

The authors considered all my comments and accordingly revised the manuscript. My recommendation is to accept the manuscript.

Author Response

National Taiwan University

Graduate Institute of Photonics and Optoelectronics (GIPO)

Department of Electrical Engineering

1, Roosevelt Rd. Sec. 4, Taipei 10617, Taiwan R.O.C.

------------------------------------------------------------------------------------------------

Dear Editor:

We would like to submit the revised manuscript (Article Number: nanomaterials-2337016.R1) entitled “Synthesis of Nano structured Ge as Transmissive or Reflective Saturable Absorber for Mode locked Fiber Laser Laser” for your consideration of publication in Nanomaterials. This manuscript has been carefully revised according to the reviewer’s comments in the separated files. Please allow us the opportunity to consider the manuscript for publication. We sincerely appreciate your kind help.

Best regards,

Gong-Ru Lin

=========================================

Professor, FIEEE, FOSA, FSPIE, FIET, FInstP

Chairman, Graduate Institute of Photonics and Optoelectronics

Department of Electrical Engineering

National Taiwan University

1, Sec. 4, Roosevelt Rd., Taipei 10617, Taiwan R.O.C.

E-mail: grlin@ntu.edu.tw

Phone: +886-2-33663700 ext. 6519

Fax: +886-2-33669598

Reviewer 2 Report

The authors properly revised the manuscript in agreement with the referee’s comments/suggestions.
There is one more thing that needs attention. This is the reference [40], which I suggest to be removed because it is incorrect. As widely demonstrated (for instance see the paper J. Phys. D: Appl. Phys. 45 (2012) 392001), the  Ref. 40 reports clean As without Si.
The first compelling synthesis of silicene was with the paper by Vogt et al  2012 Phys. Rev. Lett. 108, 155501.
Minor revision is required.

Author Response

(The authors gave the same response as above.)
